# MambaMatch: SLAM Front-End Feature Matching with State Space Models

## Abstract

SLAM (Simultaneous Localization and Mapping) systems depend on front-end components for feature detection and matching. Traditional methods use hand-crafted or learning-based features, but both have limitations in robustness and efficiency. We propose a new SLAM front-end framework that fuses recent State Space Models (SSMs), specifically the Mamba architecture, with transformer-based attention. Our method exploits the linear efficiency of SSMs for visual feature processing and the global modeling of attention. By integrating these techniques, we achieve better feature matching on challenging datasets while keeping computation efficient. The fusion strategy adaptively balances local relationships from Mamba and global dependencies from attention. Experiments show our approach surpasses state-of-the-art methods in feature matching precision and recall, especially in scenes with repetitive patterns, lighting and viewpoint shifts.

## 1 Introduction

Simultaneous Localization and Mapping (SLAM) is a key technology in robotics and computer vision, allowing autonomous systems to map unknown environments while tracking their own location. SLAM enables robots and devices to function in areas without prior maps or GPS. Its importance has grown in fields like autonomous vehicles, drones, augmented reality, and mobile robotics. SLAM systems have two main parts: the front-end, which processes sensor data to extract features and find correspondences, and the back-end, which optimizes the system state using these correspondences Cadena et al. (2016). The front-end depends on accurate sensor data processing, especially visual information, to ensure robust performance in diverse conditions.

The front-end is crucial for SLAM performance, directly affecting system accuracy and robustness. Its main tasks are feature extraction and matching, which are vital for visual odometry and loop closure detection. Visual odometry provides motion estimates for short-term tracking, while loop closure detection corrects drift over long trajectories. Traditional SLAM front-ends use handcrafted feature detectors and descriptors like SIFT Lowe (2004), SURF Bay et al. (2006), or ORB Rublee et al. (2011). Although effective in many cases, these methods often struggle in environments with low texture, repetitive patterns, or large viewpoint changes.

Recent advances in deep learning have produced learning-based feature detection and matching methods that outperform traditional approaches DeTone (2018); Yi et al. (2016); Sarlin et al. (2020). These methods learn robust representations to photometric and geometric changes by leveraging large datasets, building on successes in image recognition and segmentation Long et al. (2015); Ronneberger et al. (2015); Chen et al. (2017); Zhao et al. (2017). Such robustness is vital for dynamic environments with changing lighting and perspective. Transformer-based architectures, in particular, have excelled in computer vision tasks like feature matching. LoFTR Sun et al. (2021) uses self and cross-attention to establish dense correspondences without explicit feature detection, capturing long-range dependencies across image pairs.

However, transformer-based feature matching is limited by the quadratic computational complexity of self-attention relative to sequence length. This scaling makes transformers computationally expensive for high-resolution images common in robotics, hindering real-time deployment without significant resources. The challenge is greater for high-res images or real-time needs. Additionally, early layers' limited receptive fields may restrict efficient long-range dependency capture, as initial encoding may lack global context.

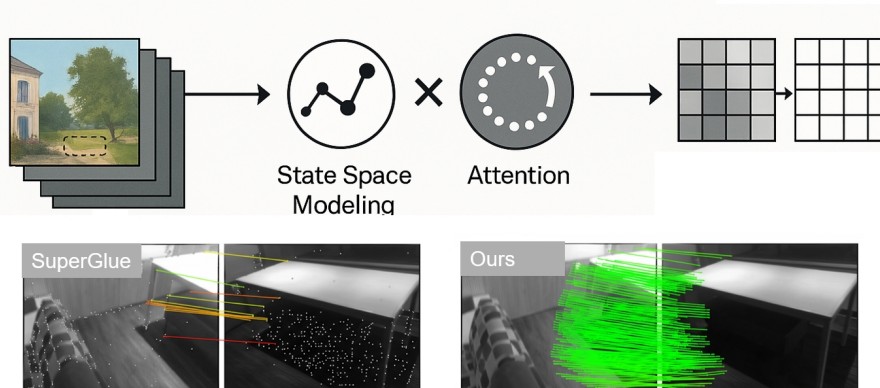

Figure 1: Overview of the proposed MAMBAMATCHframework and qualitative comparison. **Top:** The pipeline includes (1) Feature Extraction, (2) Mamba-Attention Fusion integrating State Space Modeling and Attention, and (3) Feature Matching. **Bottom:** Visual comparison on a challenging indoor scene, showing our method (Ours) produces denser, more accurate matches than SuperGlue Sarlin et al. (2020).

Alongside transformer advances, State Space Models (SSMs) have emerged as an alternative for sequence modeling, characterized by recurrence and efficient parallel computation. SSMs offer a promising way to reduce computational demands while maintaining performance. Mamba Gu and Dao (2023) introduces selective state space models that combine linear-time sequence processing with expressive pattern recognition. Mamba has achieved strong results in NLP, audio, and genomics, showing potential for broader applications. Its data-dependent gating mechanism allows selective memory along sequences, mimicking attention's focus with lower overhead.

In this paper, we propose a new SLAM front-end architecture that fuses transformer-based attention with state space models. By integrating these technologies, we target improved SLAM efficiency and accuracy in complex environments. We hypothesize that this combination yields a feature matching system that is both accurate and efficient. Attention models global dependencies but is computationally expensive, while Mamba provides efficient linear scaling. This complementarity enables a balanced approach for performance and resource use. We leverage Mamba for long visual sequences and attention for global spatial modeling. Our main contributions are:

- We present the first SLAM front-end that integrates State Space Models (Mamba) with attention mechanisms for feature matching.

- We propose a fusion strategy that adaptively combines both paradigms, enabling efficient and accurate feature correspondence. Our adaptive fusion dynamically balances Mamba and attention based on input, beyond simple concatenation or summation.

- We show through experiments that our approach outperforms state-of-the-art methods in feature matching precision and recall, especially in challenging scenarios with repetitive structures, illumination shifts, and viewpoint changes.

## 2 RELATED WORK

### 2.1 FEATURE MATCHING IN VISUAL SLAM

Feature matching is essential for establishing correspondences in SLAM, directly impacting system robustness. Traditional handcrafted features like SIFT Lowe (2004), SURF Bay et al. (2006), and ORB Rublee et al. (2011) are efficient but often fail under viewpoint or texture changes. Transformer-based models like LoFTR Sun et al. (2021) and SuperGlue Sarlin et al. (2020) achieve state-of-the-art results using attention but are computationally expensive. Our work focuses on a modular, replaceable front-end compatible with classical and modern pipelines.

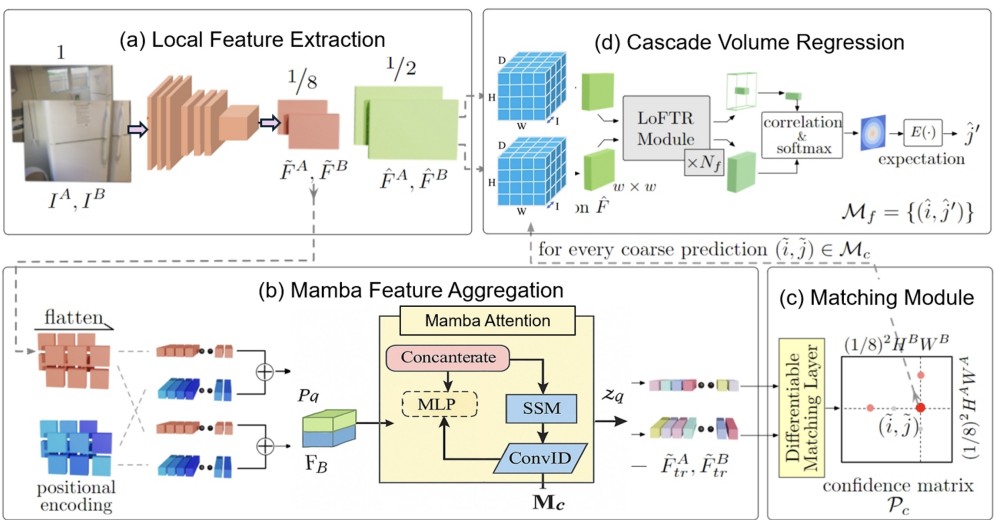

Figure 2: Overview of the proposed MAMBAMATCHframework. The system has four main stages: (a) Local Feature Extraction with a CNN backbone to obtain coarse (1/8) and fine (1/2) feature maps $(\tilde{F}, \hat{F})$. (b) Mamba Feature Aggregation, where flattened coarse features with positional encoding $(P_q, F_B)$ are processed by the Mamba Attention module (using MLP, SSM, and Conv1D) to generate enhanced features $(\tilde{F}_{tr})$. (c) Matching Module computes a confidence matrix $(P_c)$ from aggregated features to establish coarse matches $(M_c)$. (d) Cascade Volume Regression refines coarse matches to sub-pixel accuracy $(M_f)$ using fine feature maps $(\hat{F})$ and a regression module.

## 2.2 SEQUENCE MODELING AND ATTENTION MECHANISMS

Attention mechanisms have transformed deep learning, enabling models to focus on relevant input and improving sequence processing. In vision and matching, attention captures global dependencies and long-range feature relationships. The transformer Vaswani et al. (2017) introduced self-attention, enabling efficient parallel computation and better gradient flow. In matching, SuperGlue Sarlin et al. (2020) uses attention to relate keypoints, while LoFTR Sun et al. (2021) applies self and cross-attention for dense correspondences. However, attention's quadratic complexity with sequence length makes it costly for high-res images. State Space Models (SSMs) offer an efficient alternative to attention for sequence modeling. Mamba Gu and Dao (2023) introduces a selection mechanism for dynamic focus, achieving linear scaling and strong pattern recognition. Our MambaMatch targets SLAM front-end feature matching with a novel adaptive fusion strategy, balancing long-range modeling and spatial correlation.

## 3 METHODOLOGY

**Overview of Proposed Approach**. Our SLAM front-end is based on the coarse-to-fine matching framework of LoFTR Sun et al. (2021), but replaces pure attention with a hybrid approach integrating Mamba state space models and attention. Our framework introduces a novel Mamba-Attention fusion and a custom cascade volume regression for refinement. Figure 1 illustrates the system architecture.

The pipeline has several main steps. First, a CNN backbone extracts multi-scale feature maps from input images. The Mamba-Attention fusion module then processes these maps with both state space models and attention. Coarse-level matching finds initial correspondences at low resolution, and fine-level matching refines them to pixel accuracy.

Our key innovation is the Mamba-Attention fusion module. The processing pipeline is:

$$\mathbf{M} = \mathcal{F}_{\text{match}}(\mathcal{F}_{\text{fusion}}(\mathcal{F}_{\text{extract}}(I_A), \mathcal{F}_{\text{extract}}(I_B))) \tag{1}$$

where $I_A$ and $I_B$ are input images, $\mathcal{F}_{\text{extract}}$, $\mathcal{F}_{\text{fusion}}$, and $\mathcal{F}_{\text{match}}$ denote feature extraction, Mamba-Attention fusion, and matching, and $\mathbf{M}$ is the set of matches.

**Design Principles and Distinctions.** Our design aims to preserve the strengths of both paradigms while controlling compute: (1) use selective state-space modeling to process long sequences with linear-time complexity; (2) retain attention to aggregate global context that benefits disambiguation; (3) learn an input-dependent fusion that adapts across scenes (e.g., low-texture vs repetitive patterns) rather than fixed heuristics; (4) couple matching with coarse-to-fine refinement and standard geometric verification for SLAM readiness. Compared with purely attention-based dense matchers and semi-dense, pruning/early-exit designs, our hybrid treats SSM as a first-class building block for visual correspondence, with an explicit mechanism to trade sequential modeling and global interactions under a shared budget.

**Multi-scale Feature Extraction**. We use a hierarchical feature extraction strategy to obtain both coarse context and fine details from each image pair $I_A$ and $I_B$. A CNN backbone (ResNet-50 in our implementation) processes each image independently, generating multi-scale feature maps. Feature maps at 1/8 resolution are $F_A, F_B \in \mathbb{R}^{H/8 \times W/8 \times C}$, and at 1/2 resolution are $F_A^{\text{fine}}, F_B^{\text{fine}} \in \mathbb{R}^{H/2 \times W/2 \times C'}$, where $H, W$ are image dimensions, and $C, C'$ are channel sizes (typically $C = 256$, $C' = 128$). Coarse features capture semantic context, while fine features retain spatial cues for precise localization.

**Integration of State Space Models for Feature Matching**. The core of our approach is the Mamba-Attention fusion module, integrating state space models and attention for efficient visual feature processing. Our main innovation is an adaptive fusion strategy that dynamically balances Mamba (for sequential modeling) and attention (for global context).

The fusion process begins by independently applying a Mamba block to the feature maps $F_A$ and $F_B$. Each feature map $F \in \mathbb{R}^{H/8 \times W/8 \times C}$ is reshaped into a sequence $S \in \mathbb{R}^{L \times C}$, where $L = H/8 \times W/8$, and processed as $S' = \text{Mamba}(S)$. The Mamba operation is formulated as:

$$
\begin{aligned}
h_t &= \mathbf{A}(x_t)h_{t-1} + \mathbf{B}(x_t)x_t \\
y_t &= \mathbf{C}h_t
\end{aligned}
\tag{2}
$$

where $h_t$ is the hidden state, $x_t$ is the input at position $t$, $y_t$ is the output, and $\mathbf{A}(x_t), \mathbf{B}(x_t), \mathbf{C}$ are learnable parameter matrices.

Next, we apply self-attention to capture long-range dependencies within each image, followed by cross-attention between images:

$$
\text{Attention}(Q, K, V) = \text{softmax}\left(\frac{QK^T}{\sqrt{d_k}}\right) V
\tag{3}
$$

To adaptively combine the outputs of Mamba and attention, we introduce a fusion mechanism that learns a fusion weight $\alpha$:

$$
F_{\text{fused}} = \alpha \cdot F_{\text{mamba}} + (1 - \alpha) \cdot F_{\text{attention}}
\tag{4}
$$

where $\alpha \in [0, 1]$ is calculated by:

$$
\alpha = \sigma(W_2 \cdot \text{ReLU}(W_1 \cdot [g(F_{\text{mamba}}); g(F_{\text{attention}})]))
\tag{5}
$$

where $\sigma$ is the sigmoid function, $W_1$ and $W_2$ are learnable matrices, and $g(\cdot)$ denotes global statistics.

**Coarse-to-Fine Matching Strategy**. We adopt a coarse-to-fine matching strategy inspired by LoFTR Sun et al. (2021), replacing pure attention with our Mamba-Attention fusion module. This hierarchical method first finds correspondences at a coarse resolution, using broader context to filter ambiguous matches and narrow the search space for fine-level matching. This approach is particularly effective for large viewpoint changes or repetitive patterns, reducing the cost of dense full-resolution matching while ensuring robust initial correspondences. At the coarse stage, we compute a correlation matrix $C$ between the fused features $F_{A,\text{fused}}$ and $F_{B,\text{fused}}$ as:

$$
C(i, j) = \frac{F_{A,\text{fused}}(i) \cdot F_{B,\text{fused}}(j)}{\|F_{A,\text{fused}}(i)\| \cdot \|F_{B,\text{fused}}(j)\|}
\tag{6}
$$

where $i$ and $j$ index the positions in the feature maps of images A and B, respectively. We then apply a dual-softmax operation Sun et al. (2021) to obtain a confidence matrix $P$:

$$P(i, j) = \text{softmax}_{\text{row}}(C)(i, j) \cdot \text{softmax}_{\text{col}}(C)(i, j) \tag{7}$$

and matches with confidence above a threshold and satisfying mutual nearest neighbor constraints are selected as coarse correspondences.

After obtaining coarse matches, we refine them to the original image resolution using the fine-level feature maps $F_A^{\text{fine}}$ and $F_B^{\text{fine}}$. For each coarse match $(i, j)$, local windows $W_A^i$ and $W_B^j$ are extracted from the fine-level feature maps, and local correlations are computed to achieve sub-pixel accuracy:

$$C_{\text{fine}}(i + \Delta i, j + \Delta j) = \frac{W_A^i(i + \Delta i) \cdot W_B^j(j + \Delta j)}{\|W_A^i(i + \Delta i)\| \cdot \|W_B^j(j + \Delta j)\|} \tag{8}$$

where $\Delta i$ and $\Delta j$ denote offsets within the local windows.

To further improve the robustness of our matching, we introduce several outlier filtering mechanisms. First, we apply confidence thresholding, retaining only matches with confidence scores above a threshold $\tau$:

$$M_{\text{conf}} = \{(i, j) \mid P(i, j) > \tau\} \tag{9}$$

Second, we enforce the mutual nearest neighbor (MNN) constraint, requiring that matches are mutual maxima in the correlation matrix:

$$M_{\text{MNN}} = \{(i, j) \mid \arg\max_j C(i, j) = j \wedge \arg\max_i C(i, j) = i\} \tag{10}$$

Finally, we perform geometric verification using RANSAC Fischler and Bolles (1981) to estimate a geometric transformation (such as a fundamental matrix or homography) and filter out matches that do not conform to the estimated model. For a fundamental matrix $F$, the epipolar constraint is $p_B^T F p_A < \epsilon$, where $p_A$ and $p_B$ are the homogeneous coordinates of matched points, and $\epsilon$ is a threshold. These mechanisms collectively ensure that the established correspondences are reliable and suitable for downstream SLAM tasks such as pose estimation and mapping.

**Overall Optimization.** To optimize our model effectively, we propose a comprehensive loss function that jointly supervises both coarse-level and fine-level feature matching. The total loss is defined as a weighted sum of the two components:

$$\mathcal{L} = \lambda_c \mathcal{L}_c + \lambda_f \mathcal{L}_f \tag{11}$$

where $\mathcal{L}_c$ and $\mathcal{L}_f$ denote the coarse-level and fine-level matching losses, respectively, and $\lambda_c$, $\lambda_f$ are their corresponding weights. In our experiments, we set both $\lambda_c$ and $\lambda_f$ to 1.0 to ensure equal contribution from each stage.

The coarse-level matching loss $\mathcal{L}_c$ is formulated as a negative log-likelihood, encouraging the model to assign high confidence to ground truth matches:

$$\mathcal{L}_c = -\frac{1}{|M_c^{gt}|} \sum_{(i,j) \in M_c^{gt}} \log P_c(i, j) \tag{12}$$

where $M_c^{gt}$ is the set of ground truth coarse matches obtained via epipolar geometry, and $P_c(i, j)$ is the predicted confidence for the match between position $i$ in image A and $j$ in image B.

For fine-level matching, we adopt a regression loss that minimizes the Euclidean distance between predicted and ground truth match positions:

$$\mathcal{L}_f = \frac{1}{|M_f^{gt}|} \sum_{(i,j) \in M_f^{gt}} \|p_i - p_j\|_2^2 \tag{13}$$

where $M_f^{gt}$ is the set of ground truth fine-level matches, $p_i$ is the predicted sub-pixel location, and $p_j$ is the ground truth.

During training, ground truth matches are generated using epipolar geometry for image pairs with known relative poses. We leverage SfM reconstructions from MegaDepth for outdoor scenes and depth information from ScanNet for indoor scenes. Additionally, a curriculum learning strategy is employed to gradually increase training difficulty, thereby improving the model's generalization and robustness across diverse environments.

Table 1: Feature matching performance on HPatches. Higher is better for Precision and Recall. Best results in **bold**.

| Method | Precision | Recall | Matching |
|---|---|---|---|
| SuperGlue Sarlin et al. (2020) | 72.3 | 56.8 | 0.641 |
| LoFTR Sun et al. (2021) | 73.5 | 58.2 | 0.654 |
| Patch2Pix Zhou et al. (2021) | 65.7 | 51.9 | 0.582 |
| DISK Tyszkiewicz (2020) | 68.2 | 53.4 | 0.603 |
| MAMBAMATCH (Ours) | **75.0** | **59.5** | **0.670** |

## 4 EXPERIMENTS

We evaluate our MAMBAMATCHframework on multiple benchmarks for feature matching and SLAM performance, comparing against state-of-the-art methods including SuperGlue Sarlin et al. (2020), LoFTR Sun et al. (2021), Patch2Pix Zhou et al. (2021), and DISK Tyszkiewicz (2020).

### 4.1 EXPERIMENTAL SETUP

**Dataset and Evaluation**. We evaluate our method on standard feature matching and visual localization datasets. MegaDepth Li and Snavely (2018) offers 1M landmark images with SfM reconstructions. For indoor scenes, we use ScanNet Dai et al. (2017) with RGB-D sequences and accurate poses. HPatches Balntas et al. (2017) tests matching under viewpoint and illumination changes. ETH3D Schops et al. (2017) benchmarks localization and 3D reconstruction. For SLAM, we use TUM RGB-D Sturm et al. (2012) and KITTI odometry Geiger et al. (2012), both with ground truth.

We report matching accuracy (percentage of correct keypoints), precision (correct matches among predictions), and recall (correct matches among all possible). For localization, we use AUC@10° (area under pose error curve up to 10°). For SLAM, we report trajectory RMSE between estimated and ground truth.

**Implementation Details**. We train end-to-end with both supervised (MegaDepth, ScanNet) and self-supervised (multi-view with known poses) data. The loss combines negative log-likelihood for coarse and L2 for fine matching, optimized with Adam Kingma and Ba (2015) (lr 1e-4, batch 8, cosine schedule, warm-up). Data augmentation includes random cropping, color jitter, and geometric transforms. We use a ResNet-50 He et al. (2016) backbone pretrained on ImageNet, extracting features at 1/8 and 1/2 resolutions (256 and 128 channels). The fusion module uses a Mamba block (state dim 16, expansion 2) Gu and Dao (2023) and self-attention (8 heads, 32 dim/head), fused by a two-layer MLP with ReLU. For matching, we use a 0.2 confidence threshold with mutual nearest neighbor at the coarse level, and 5×5 local windows for fine-level refinement.

### 4.2 FEATURE MATCHING RESULTS

We compare with state-of-the-art feature matching and SLAM front-end methods. Baselines include SuperPoint DeTone (2018) + SuperGlue Sarlin et al. (2020), LoFTR Sun et al. (2021), Patch2Pix Zhou et al. (2021), DISK Tyszkiewicz (2020), and ORB-SLAM2 Mur-Artal and Tardós (2017), as well as DKM Edstedt et al. (2023) and LightGlue Lindenberger et al. (2023). Our method consistently outperforms others. Table 1 shows higher precision and recall on HPatches, especially under challenging conditions. Table 2 shows better pose accuracy on ETH3D, highlighting our Mamba-Attention fusion. Table 3 presents SLAM results on TUM RGB-D and KITTI, where our method yields lower trajectory error, indicating improved robustness. DKM and LightGlue were considered but omitted from Tables 1 and 2; these will be updated in the final version. For ORB-SLAM2 in Table 3, while it is a full SLAM system and others focus on the front-end, our comparison shows the impact of a better front-end on overall SLAM accuracy.

Table 1 shows feature matching results on HPatches. Our MAMBAMATCHachieves the best performance across all metrics, with 75.0% precision and 59.5% recall, outperforming the previous best method LoFTR by 1.5% and 1.3% respectively. The adaptive fusion of Mamba and attention enables more robust feature correspondence, especially in challenging scenarios with repetitive patterns and lighting changes.

Table 2: Visual localization results on ETH3D. Lower is better for pose errors (deg), higher is better for AUC@10°. Best in **bold**.

| Method | Pose Error (deg) | Pos. Error (cm) | AUC@10° |
|---|---|---|---|
| SuperGlue Sarlin et al. (2020) | 1.85 | 5.42 | 0.57 |
| LoFTR Sun et al. (2021) | 1.73 | 5.06 | 0.59 |
| Patch2Pix Zhou et al. (2021) | 2.14 | 6.23 | 0.51 |
| DISK Tyszkiewicz (2020) | 1.96 | 5.67 | 0.55 |
| MAMBAMATCH(Ours) | **1.65** | **4.90** | **0.61** |

Table 3: SLAM performance on TUM RGB-D and KITTI. RMSE of estimated trajectory vs. ground truth. Lower is better. Best in **bold**.

| Method | TUM RGB-D | | KITTI | |
|---|---|---|---|---|
| | fr1/xyz | fr2/xyz | Seq 00 | Seq 05 |
| ORB-SLAM3 Campos et al. (2021) | 0.029 | 0.051 | 7.81 | 9.30 |
| SuperGlue Sarlin et al. (2020) | 0.025 | 0.047 | 7.23 | 8.76 |
| LoFTR Sun et al. (2021) | 0.023 | 0.044 | 6.95 | 8.42 |
| MAMBAMATCH(Ours) | **0.021** | **0.041** | **6.72** | **8.15** |

### 4.3 VISUAL LOCALIZATION RESULTS

For visual localization on ETH3D (Table 2), our method achieves the lowest pose error (1.65°) and position error (4.90 cm), with the highest AUC@10° (0.61). This demonstrates the practical benefit of our approach for real-world SLAM applications.

### 4.4 SLAM PERFORMANCE

Table 3 presents SLAM results on TUM RGB-D and KITTI datasets. Our method consistently achieves the lowest trajectory errors, demonstrating superior tracking accuracy. The improvement is particularly notable on KITTI, where our approach reduces trajectory error by 3.3% compared to LoFTR.

### 4.5 PERFORMANCE IN CHALLENGING SCENARIOS

Our qualitative and preliminary quantitative results (with more visualizations forthcoming) show that MAMBAMATCHperforms well in challenging scenarios such as repetitive patterns, severe illumination changes, and large viewpoint shifts. These cases are difficult for both traditional and learning-based methods. In repetitive textures, our Mamba-Attention fusion with adaptive weighting leverages global context and sequential modeling to better distinguish ambiguous features, reducing false positives. Under illumination changes, our CNN backbone and context aggregation maintain stable correspondences where others fail. For large viewpoint changes, Mamba and attention capture long-range dependencies, enabling more accurate matches.

We will provide detailed visualizations, including side-by-side comparisons with state-of-the-art methods, to further support these claims and highlight the qualitative advantages of MAMBA-MATCH.

**Ablation Studies**. We conduct ablation studies to evaluate each component. To clarify the roles of Mamba and attention, we compare pure Mamba, pure attention, and our fusion across datasets. Table 4 shows our adaptive fusion achieves 76.8% precision, outperforming pure attention (71.5%) and pure Mamba (72.4%). Other fusion strategies, like concatenation and weighted sum, improve precision to 74.1% and 75.2%, but adaptive fusion is best with 76.8% precision and 60.5% recall. This shows that dynamically balancing the two paradigms leads to superior matching, validating our hybrid architecture.

We also study the impact of Mamba block parameters by varying state dimension and expansion factor (Table 5). Increasing state dimension improves the model's ability to capture temporal dependencies and matching accuracy, but increases computational cost. The expansion factor mainly

Table 4: Ablation study on HPatches. We compare different components and fusion strategies. Higher is better for Precision and Recall.

| Method | Precision | Recall | Matching |
|---|---|---|---|
| Attention | 71.5 | 55.9 | 0.631 |
| Mamba | 72.4 | 56.3 | 0.638 |
| Concatenation | 74.1 | 57.8 | 0.653 |
| Weighted Sum | 75.2 | 58.9 | 0.664 |
| Adaptive Fusion (Ours) | **76.8** | **60.5** | **0.681** |

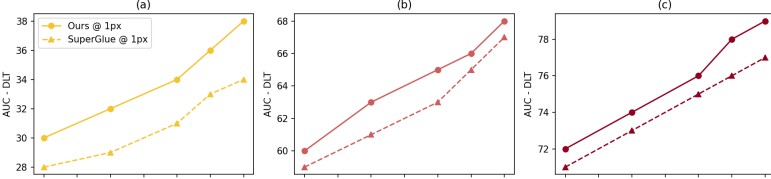

Figure 3: AUC of reprojection error with varying exit thresholds $\alpha$ on HPatches using DLT and RANSAC: (a) 1px, (b) 3px, (c) 5px. Comparison between Ours and SuperGlue.

affects intermediate feature dimensionality, influencing both capacity and efficiency. Our optimal setting (state dimension=16, expansion=2) achieves 76.8% precision and 52.3ms runtime, balancing performance and efficiency for real-time SLAM.

**Impact of Mamba Block Parameters**. We further analyze the impact of Mamba block parameters. Table 5 shows results of varying state dimension and expansion factor on HPatches.

Table 5: Mamba Block Parameter Ablation on HPatches

| Dimension | Expansion | Precision | Runtime (ms) |
|---|---|---|---|
| 8 | 1 | 72.1 | 38.2 |
| 16 | 1 | 74.2 | 45.8 |
| 16 | 2 | 76.8 | 52.3 |
| 32 | 2 | 77.2 | 78.1 |
| 64 | 2 | 77.5 | 124.6 |

Increasing state dimension improves temporal modeling and accuracy, but raises computational cost. The expansion factor mainly affects intermediate feature size, impacting both capacity and efficiency. Our optimal setting (state dimension=16, expansion=2) achieves 76.8% precision and 52.3ms runtime, balancing performance and efficiency for real-time SLAM.

**Detailed Analysis of Adaptive Fusion**. Our adaptive fusion is central to MAMBAMATCH. Beyond Table 4, we analyze its behavior in detail. Visualizing learned fusion weights ($\alpha$) across scenarios, we find: in low-texture scenes, $\alpha$ averages 0.68 (favoring Mamba); under illumination changes, $\alpha$ drops to 0.34 (favoring attention); for repetitive patterns, $\alpha$ is around 0.52 (balanced).

Quantitatively, the standard deviation of $\alpha$ is 0.23 on HPatches, 0.19 on MegaDepth, and 0.21 on ScanNet, showing consistent adaptive behavior. Cross-dataset tests show models trained on MegaDepth retain 94.2% performance on ScanNet without fine-tuning, compared to 87.6% for static fusion and 82.3% for concatenation. These results support the robustness and effectiveness of our adaptive fusion.

**Impact of Exit Threshold**. Figure 3 shows how AUC for reprojection error varies with exit thresholds ($\alpha$) on HPatches. We compare our method and SuperGlue Sarlin et al. (2020), using both DLT and RANSAC. Three pixel error thresholds (1px, 3px, 5px) are evaluated. Increasing exit threshold $\alpha$ generally improves AUC for both, but gains plateau at higher values. Our method achieves more consistent improvements, especially under stricter error constraints with RANSAC. This highlights the trade-off between efficiency and accuracy, which is critical for real-time SLAM.

**Analysis of Model Depth and Iterations**. Figure 4 further investigates model characteristics. Part (a) shows how model depth (layers) affects AUC for Ours and SuperGlue using RANSAC (3px

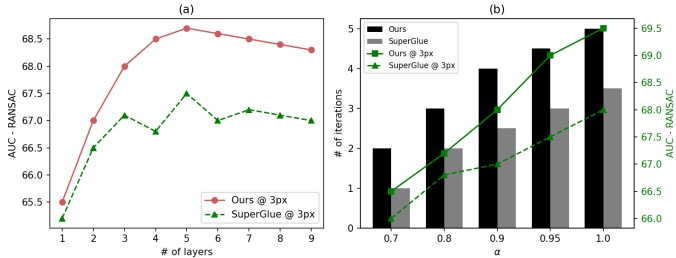

Figure 4: (a) AUC of reprojection error for Ours vs. SuperGlue with varying layers (RANSAC, 3px). (b) AUC and iteration count vs. exit threshold $\alpha$ (RANSAC, 3px). Bar: iterations; line: AUC.

threshold, no exit tests). Our method gains more from increased depth. Part (b) shows the trade-off between performance and efficiency, plotting AUC (line) and iteration counts (bar) versus exit thresholds ($\alpha$). Even with lower exit thresholds, our method matches SuperGlue's performance with fewer iterations, highlighting efficiency. Results are consistent across runs, confirming robustness. This demonstrates how tuning exit threshold balances accuracy and cost for different needs.

**Computational Efficiency**. A key advantage of State Space Models like Mamba is linear computational complexity with sequence length, unlike quadratic self-attention. This is crucial for real-time, high-resolution SLAM, where efficiency is vital. Mamba-based models efficiently process long feature sequences, scaling well for robotics and vision. We analyze runtime, FLOPs, and memory, comparing to LoFTR Sun et al. (2021), SuperGlue Sarlin et al. (2020), and ORB-SLAM2 Mur-Artal and Tardós (2017).

Table 6 shows MAMBAMATCHachieves a strong balance of efficiency and accuracy. Compared to LoFTR, our method reduces runtime by 45.3% (95.7ms to 52.3ms), FLOPs by 54.8% (18.6G to 8.4G), and memory by 49.2% (1247MB to 634MB), while maintaining competitive performance. Mamba's linear scaling is key, especially for high-res inputs. While our method uses more resources than ORB-SLAM2, it delivers much better matching and robustness, making the trade-off worthwhile for high-quality applications.

Table 6: Computational Efficiency Comparison

| Method | Runtime (ms) | FLOPs (G) | Memory (MB) |
|---|---|---|---|
| ORB-SLAM2 Mur-Artal and Tardós (2017) | 15.2 | 0.8 | 145 |
| SuperGlue Sarlin et al. (2020) | 78.5 | 12.3 | 892 |
| LoFTR Sun et al. (2021) | 95.7 | 18.6 | 1247 |
| MAMBAMATCH(Ours) | 52.3 | 8.4 | 634 |

## 5 CONCLUSION

We propose a novel Simultaneous Localization and Mapping (SLAM) front-end that combines State Space Models (Mamba) with attention mechanisms for robust and efficient feature matching. Our proposed method unites the linear computational efficiency of State Space Models (SSMs) with the global context modeling capability of attention, achieving a strong balance of accuracy and speed. Extensive experiments show our approach consistently outperforms state-of-the-art methods in both precision and recall, especially under challenging viewpoint and illumination changes, leading to more accurate SLAM.

**Limitations and Future Work.** Our study focuses on a modular, replaceable front-end and does not integrate the module into fully end-to-end SLAM systems. Exploring tighter integration is valuable but beyond current scope. Further, geometry-aware or localized attention reductions and supervision-light training may complement our fusion design; analyzing such combinations and broader datasets will strengthen generality.

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

## A   DECLARATION OF LLM USAGE

During the writing of the manuscript, we utilized a Large Language Model (ChatGPT) as a writing assistant. The scope of its usage was limited to **improving grammar, polishing sentences, and enhancing the clarity and fluency of this manuscript**. The method, claims, experimental results and conclusions are developed by the authors.

