# OpenReview forum: "MambaMatch: SLAM Front-End Feature Matching with State Space Models"
_ICLR.cc/2026/Conference — ICLR 2026 Conference Withdrawn Submission_

### Official Review · Reviewer_WTEe · 2025-10-27

**Soundness:** 1
**Presentation:** 2
**Contribution:** 1
**Rating:** 2
**Confidence:** 5

**Summary:**

This paper proposes a semi-dense matcher, where the main novelty lies in combining Mamba and Transformer for feature extraction. However, it’s not clear to me why this design is specifically tailored for the SLAM front-end. Some statements are not very precise, and the comparative methods used are somewhat outdated.

**Strengths:**

The paper is easy to follow.

**Weaknesses:**

It’s not clear why the authors emphasize that the method is tailored for the SLAM front-end. The related work section seems incomplete, suggesting that the literature has not been fully reviewed. Overall, the contribution and evaluation do not yet reach the standard of a top-tier conference such as ICLR.

**Questions:**

See weaknesses.

---

### Official Review · Reviewer_MkH5 · 2025-10-29

**Soundness:** 2
**Presentation:** 3
**Contribution:** 2
**Rating:** 4
**Confidence:** 4

**Summary:**

The paper introduces a novel method for feature matching between pairs of images. It builds upon the coarse-to-fine architectural design proposed in [1], but differs by incorporating a fusion of MAMBA state-space modeling and the attention mechanism to process features at the coarse level. In addition, the proposed model learns an input-dependent weighting factor to adaptively balance the contributions of the two fused components.

[1] Jiaming Sun, Zehong Shen, Yuang Wang, Hujun Bao, and Xiaowei Zhou. LoFTR: Detector-free local feature matching with transformers. arXiv preprint arXiv:2104.00680, 2021.

**Strengths:**

The paper effectively combines MAMBA and attention to exploit complementary context modeling strategies: attention captures fine-grained spatial relationships, while MAMBA provides smooth, coherent state representations that stabilize long-range dependencies. The input-dependent fusion adaptively balances these components, improving the robustness of coarse-level descriptors and reducing false correspondences in low-texture or ambiguous regions. The learned weighting further enhances generalization by dynamically adjusting between global smoothness and local distinctiveness.

**Weaknesses:**

The authors claim that their method improves both the efficiency and accuracy of feature matching compared to approaches relying solely on attention. However, although MAMBA is theoretically more efficient than attention due to its linear-time state updates, the proposed fusion module computes both the attention and MAMBA branches in parallel and combines their outputs through an input-dependent weighting. As a result, the overall computational complexity remains dominated by the quadratic cost of attention, negating the intended efficiency advantage. In practice, the model may even experience higher inference latency and memory usage due to duplicated computations and additional fusion parameters.

The overall pipeline, including the coarse-to-fine refinement, correspondence formation, and positional encoding strategy, largely mirrors LoFTR. The primary modification is the replacement of the coarse-level transformer with a MAMBA–attention fusion block. While this hybridization is conceptually interesting, it may be viewed as an incremental architectural adjustment rather than a fundamentally new paradigm.

**Questions:**

- In Equation (5), g(⋅) denotes global statistics. Could the authors specify which global statistics were used?

- In the experimental section, the authors compare their proposed fusion strategy with a baseline that performs a weighted sum between MAMBA and attention features. However, the distinction between the two approaches is not clearly explained. Does the baseline use a fixed weighting coefficient rather than a learned one? If so, was any hyperparameter search or tuning performed to determine the optimal static weight?

- Since the proposed MAMBA component is causal along a chosen one-dimensional ordering of the two-dimensional feature map, the method may inherit a directional bias. A row-major sequence emphasizes horizontal context while underutilizing vertical dependencies, and vice versa for a column-major sequence. This may reduce rotation robustness and produce anisotropic features. It would useful to conduct an ablation study across different sequence orderings (row, column, snake, or space-filling curves such as Hilbert and Morton), as well as testing bidirectional and multi-axis SSM variants and evaluating rotation and flip robustness. Visualizing the fusion weights and feature spectra would also help clarify whether the attention branch compensates for this bias and whether order-invariant designs (e.g., multi-axis or bidirectional SSM, or order augmentation) are necessary.

- Finally, could the authors justify how their approach achieves better computational efficiency than LoFTR, given that it employs both MAMBA and attention while LoFTR uses only attention in the coarse-level block?

---

### Official Review · Reviewer_NG27 · 2025-10-31

**Soundness:** 1
**Presentation:** 1
**Contribution:** 1
**Rating:** 2
**Confidence:** 3

**Summary:**

The paper introduces a image matching method that combines State Space Models (SSMs)—specifically the Mamba architecture—with transformer-style attention.

The SSM branch delivers linear-time visual feature processing, while the attention branch supplies global context. An adaptive fusion module balances the local correlations captured by Mamba with the long-range dependencies modeled by attention.

Experiments demonstrate higher feature-matching precision and recall.

**Strengths:**

The paper merge a State Space Model (Mamba) with transformer attention in a SLAM front-end, offering a way to exploit their complementary properties (linear efficiency vs. global context).

The proposed adaptive fusion goes beyond naive concatenation, balancing the two pathways according to the input. The experimental protocol shows gains in both precision and recall.

Achieving superior feature matching in scenes with repetitive patterns, illumination changes, and viewpoint shifts addresses a long-standing challenge in SLAM.

**Weaknesses:**

Figure 1 is not informative: it is almost identical to LoFTR, and some text is obscured by the images.

Figure 2 is unclear and again closely resembles LoFTR; the figure should be revised to highlight the new contributions.

The Method section repeats much of LoFTR; please separate preliminaries from your own contributions more explicitly.

**Questions:**

Please provide qualitative visualizations for both the matching and the SLAM results.

The current narrative blurs the line between a image matcher and a SLAM‐specific contribution. As the work essentially replaces the SLAM matching module with a “LoFTR + Mamba” variant, would it be clearer to reframe the paper around feature matching with an application to SLAM?

The manuscript lacks discussion of closely related work, notably

    MambaGlue: Fast and Robust Local Feature Matching with Mamba
    JamMa: Ultra-lightweight Local Feature Matching with Joint Mamba

Please include quantitative comparisons against the following strong baselines:

    Efficient LoFTR: Semi-Dense Local Feature Matching with Sparse-Like Speed
    MatchAnything: Universal Cross-Modality Image Matching with Large-Scale Pre-Training
    RoMa: Robust Dense Feature Matching
    UFM: A Simple Path towards Unified Dense Correspondence with Flow

---

### Official Review · Reviewer_1S91 · 2025-11-07

**Soundness:** 3
**Presentation:** 2
**Contribution:** 3
**Rating:** 2
**Confidence:** 5

**Summary:**

See Questions

**Strengths:**

See Questions

**Weaknesses:**

See Questions

**Questions:**

After reading the paper, I have the following comments. I hope the authors could address them carefully.

- Q1. The abstract does not explain the necessity of using Mamba.

- Q2. Figure 1 contains very limited information — it's unclear what the authors intend to convey. Figure 2 is low-resolution and not a vector graphic.

- Q3. As a work that involves both SLAM and Transformer/Mamba fields, the number of references is extremely limited — only 29 papers are cited. This is not up to the standard of a top-tier conference paper.

- Q4. The authors should illustrate the process from Equation (2) to Equation (5) in a diagram. The operation flow is hard to align with what’s shown in Figure 2.

- Q5. The equations in this paper are poorly presented. For example, using capital letters A and B as subscripts is inappropriate. Also, A and B are used for image pairs, while C is a relation matrix — this inconsistency reflects a lack of clarity and indicates that the authors may still lack experience in research and academic writing.

- Q6. The work is somewhat engineering-oriented. The main content is just a combination of CNN, Mamba, MNN, RANSAC, etc. The originality of the method appears weak.

- Q7. The experiments are too limited. For example, on the TUM RGB-D and KITTI datasets, the authors only test on 4 sequences, while KITTI contains more than 10 sequences. The method should also be comprehensively compared with classical methods such as ORB-SLAM3. Moreover, visualized trajectory plots should be included to demonstrate the localization performance.

- Q8. The overall workload of the paper seems too small. There is almost no supplementary material. As the first SLAM work combining Mamba, there should logically be a larger set of experiments. From the perspective of contribution volume, this paper does not meet the bar for a top-tier venue like ICLR.

- Q9. How is the proposed method implemented in code? I care about whether the code will be open-sourced. Can the implementation stand up to peer verification?

- Q10. Some minor issues: (a) There are missing spaces after method names in lines 282 and 320.

- **Overall**, the motivation of this paper is promising, and combining Mamba with SLAM is indeed meaningful. To some extent, I would be willing to accept this paper. However, considering the above critical issues, I currently recommend rejection. I will reconsider changing my rating based on the authors’ rebuttal.

**Details Of Ethics Concerns:**

See Questions

---

### Note · Authors · 2025-11-12

I have read and agree with the venue's withdrawal policy on behalf of myself and my co-authors.